# The Potyviruses: An Evolutionary Synthesis Is Emerging

**DOI:** 10.3390/v12020132

**Published:** 2020-01-22

**Authors:** Adrian J. Gibbs, Mohammad Hajizadeh, Kazusato Ohshima, Roger A.C. Jones

**Affiliations:** 1Emeritus Faculty, Australian National University, Canberra, ACT 2601, Australia; 2Department of Plant Protection, Faculty of Agriculture, University of Kurdistan, P.O. Box 416, Sanandaj, Iran; 3Laboratory of Plant Virology, Department of Applied Biological Sciences, Faculty of Agriculture, Saga University, 1-banchi, Honjo-machi, Saga 840-8502, Japan; ohshimak@cc.saga-u.ac.jp; 4The United Graduate School of Agricultural Sciences, Kagoshima University, 1-21-2410 Korimoto, Kagoshima 890-0065, Japan; 5Institute of Agriculture, University of Western Australia, 35 Stirling Highway, Crawley, WA 6009, Australia

**Keywords:** potyviruses, potyvirids, phylogenetics, population genetics, evolution, recombination, speciation, metagenomes

## Abstract

In this review, encouraged by the dictum of Theodosius Dobzhansky that “Nothing in biology makes sense except in the light of evolution”, we outline the likely evolutionary pathways that have resulted in the observed similarities and differences of the extant molecules, biology, distribution, etc. of the potyvirids and, especially, its largest genus, the potyviruses. The potyvirids are a family of plant-infecting RNA-genome viruses. They had a single polyphyletic origin, and all share at least three of their genes (i.e., the helicase region of their CI protein, the RdRp region of their NIb protein and their coat protein) with other viruses which are otherwise unrelated. Potyvirids fall into 11 genera of which the potyviruses, the largest, include more than 150 distinct viruses found worldwide. The first potyvirus probably originated 15,000–30,000 years ago, in a Eurasian grass host, by acquiring crucial changes to its coat protein and HC-Pro protein, which enabled it to be transmitted by migrating host-seeking aphids. All potyviruses are aphid-borne and, in nature, infect discreet sets of monocotyledonous or eudicotyledonous angiosperms. All potyvirus genomes are under negative selection; the HC-Pro, CP, Nia, and NIb genes are most strongly selected, and the PIPO gene least, but there are overriding virus specific differences; for example, all turnip mosaic virus genes are more strongly conserved than those of potato virus Y. Estimates of dN/dS (ω) indicate whether potyvirus populations have been evolving as one or more subpopulations and could be used to help define species boundaries. Recombinants are common in many potyvirus populations (20%–64% in five examined), but recombination seems to be an uncommon speciation mechanism as, of 149 distinct potyviruses, only two were clear recombinants. Human activities, especially trade and farming, have fostered and spread both potyviruses and their aphid vectors throughout the world, especially over the past five centuries. The world distribution of potyviruses, especially those found on islands, indicates that potyviruses may be more frequently or effectively transmitted by seed than experimental tests suggest. Only two meta-genomic potyviruses have been recorded from animal samples, and both are probably contaminants.

## 1. Historical Origins

At a meeting of the Fellows of the National Institute of Agricultural Botany in Cambridge, UK, on 14 November 1924, Dr. Redcliffe Salaman gave a lecture entitled “Degeneration of the Potato—An Urgent Problem” [1]. He reported that “potato degeneration”, namely the decrease in yield when potatoes were grown year after year from tubers, rather than from true seed, cost the UK between five and ten million pounds sterling each year. He noted that the condition was first reported in 1778 at a meeting in Manchester, and called “potato curl”. It was worse in lowland crops and in the Southern UK than in crops grown on higher ground and in the north, and although some thought it was caused by disease, perhaps insect-borne, others believed it was a form of senility resulting from repeated vegetative reproduction! Salaman concluded that that degeneration was caused by a complex of tuber-borne pathogens.

Salaman’s talk was successful, as it induced the Ministry of Agriculture to found the Potato Virus Research Station in Cambridge and appoint him as director, and in turn he appointed Kenneth Smith as entomologist, who soon separated some of the components of potato curl and identified the viruses he called potato virus X and potato virus Y (PVY) [2].

Other viruses similar to PVY were soon reported, for example, henbane mosaic virus [3], which was like PVY in causing mosaic symptoms, being transmitted by sap, although relatively unstable in it, and also by being transmitted by aphids in short feeds. These viruses, which became known as potyviruses, short for “potato virus Y group viruses” [4], were among those included in early attempts to devise biological taxonomies of plant viruses [5] based on the length of their filamentous particles [6]. They were also distinguished from other plant viruses by having serologically distinct virions and, biologically, by having distinct host ranges and causing distinct symptoms, and by their properties in infective sap, such as dilution end point, thermal inactivation point, and longevity in vitro. Sixteen different potyviruses had been described in 1959. Subsequently, in this pre-sequencing era, a combination of techniques, including sucrose density gradient centrifugation, analytical ultracentrifugation, ultraviolet spectrophotometry, and polyacrylamide gel electrophoresis were also included to establish the sedimentation coefficients and buoyant densities of virions, and the molecular weights of protein subunits and % nucleic acid contents, as all these properties provided additional distinguishing characteristics when novel viruses were being described [7].

Virus identification and taxonomy were transformed later, when methods for sequencing genes were invented in the 1970s and applied to plant viruses [8,9], and it was established that hierarchical groupings based on viral protein and gene sequences, including those of potyviruses, confirmed and extended those that had been devised previously by using phenotypic characters, serological tests, etc. As a result, 57 potyviruses had been identified by 1991 [10], using sequences of the “part NIb-CP” region of their genomes, as this was bracketed by convenient primer sites [11,12]. By 2000, over 1000 potyvirus sequences were recorded in the GenBank database, and there are now more than 26,000. The potyviruses now form a family, the *Potyviridae* [13], containing at least eight genera of which the aphid-transmitted potyviruses, including the first described, PVY, make up the largest genus, *Potyvirus*. This large plant virus genus is one of the most important economically because of the yield and quality losses it causes in a wide range of crops worldwide. Moreover, some of its members currently endanger food security in developing countries by causing devastating diseases in tropical and subtropical food crops [14].

## 2. The Origins of the Potyviridae

The potyvirids are distinguished from other viruses by specific molecular differences, together with a combination of phenotypic properties [13,15]. All potyviruses infect plants; most are transmitted in nature by arthropods—mostly by aphids—though bymoviruses are transmitted by root-infecting plasmodiophorids, which are cercozoan amoebae. Some potyviruses are seed-borne [16]. Potyvirid virions are flexuous filaments, 680–900 nm long and 11–20 nm in diameter. Each is helically constructed from 1400 to 2140 subunits of a coat protein (CP), and a positive-sense, single-stranded RNA genome (usually monopartite, but bipartite in the genus *Bymovirus*) of 8–11 kb in total, which is wound into a groove within the CP subunits.

The ancestry and origins of the potyvirids is being revealed by studies of the structure and sequences of their genes and proteins. These show that their genomes are polyphyletic in origin, as there are significant similarities between three of their genes and those of viruses in three otherwise-unrelated virus genera; two detected by the protein sequence similarity of the helicase region of the CI protein and the RdRp region of the NIb protein, respectively, and the third by the structure of the CP [17,18]. A BLASTp search of the GenBank protein sequence database (Sept 2019), using the eight main motif regions of the polyprotein of PVY (NC_001616), and excluding matches with sequences from the *Potyviridae*, found only significant matches between the “DEAD helicase-helicase C” region of the CI protein and that of classical swine fever and hog cholera pestiviruses (chance probability, 1e-12_-16; 30% identity, 13% indels), and no others. Likewise, there were significant similarities between the PVY RdRp region and that of astroviruses (see below). Structural studies have only been reported for the CP protein of one potyvirus, watermelon mosaic virus (WMV), and reveal a structure that is closely similar to those of other viruses with flexuous filamentous virions, including two potexviruses, and also the enveloped flexuous nucleoproteins of orthomyxoviruses and bunyavirids, which include tomato spotted wilt tospovirus [19,20,21]; all these CPs have a core domain rich in alpha helices. Each CP subunit interacts with five nucleotides (nts), and has 8.8 subunits per turn in a left-handed helix, with a pitch of 34.5–35 Å. Its N-terminus is external to the virion, and its C-terminus internal. The terminal regions interact with adjacent subunits and provide flexibility to the virion. In serological studies, the N-terminus is dominant and, in potyviruses, is also involved with aphid interactions [22,23]. 

The sequences of their RNA-dependent RNA polymerases (RdRps) place the potyvirids in the “Picornavirus Supergroup” (Figure 1) [19,24], where the potyvirids are outsiders, as most of the others have icosahedral virions made of eight-stranded antiparallel beta-barrel proteins, the so-called ‘jelly roll’ proteins. In the Wolf et al. [24] taxonomy of RdRps (Figure 2), the potyvirid RdRps form a cluster that is sister to an RdRp found in a metagenome, bufivirus UC1-gp2, isolated from “wastewater” collected in San Francisco. The sister clade to the potyvirid/bufivirus clade of RdRps are mostly those of the astroviruses, a group of gut-infecting viruses that are found in a wide range of animals, mostly mammals or birds. They have 28–35 nm diameter isometric virions (https://en.wikipedia.org/wiki/Astrovirus (accessed July 2019)). Sister to the RdRp clade of potyviruses/bufivirus/astroviruses are the RdRps of hypoviruses, amalgaviruses, partitiviruses, and picobirnaviruses, many of them metagenomes, including one from *Phytophthora infestans* [25] and two from leeches [26]. None of the motifs identified in potyvirus proteins, other than the RdRp, match those encoded by astroviruses; the nonstructural protein of a human astrovirus (NP_059443) was found to have the RdRp motif, but, in addition, only a trypsin-like peptidase, a restriction enzyme, and a motif of unknown function. The bufivirus metagenome includes a 3′ terminal S domain (jelly-roll) capsid protein gene indicating that it, like most of the picornavirus supergroup, including astroviruses, probably has isometric virions.

Most of the genome of all potyvirids encodes a single polyprotein, which is post-translationally hydrolyzed into ten proteins [15,27]. It also encodes another protein (P3N-PIPO) in the −1 reading frame, and a second (P1N-PISPO) in a few potyvirids [28]. From the N-terminus to the C-terminus, the ten potyvirus proteins are named as follows: P1-Pro, HC-Pro, P3, 6K1, CI, 6K2, NIa-Pro, Nib, and CP. The P1-Pro protein is a serine protease (S30) that self-hydrolyses its own C-terminal cleavage site. Next is the HC-Pro protein, which is a cysteine protease (C6) that also hydrolyses its own C-terminal cleavage site. The other eight proteins have seven cleavage sites hydrolyzed by NIa-Pro, the cysteine protease (C4), encoded by the eighth region. The eight largest motifs were found in all potyvirus sequences by using the motif-matching facility Pfam, in all three rymoviruses (agropyrum mosaic, ryegrass mosaic, and hordeum mosaic viruses), and in the two most closely related potyvirids, namely reed chlorotic stripe virus and blackberry virus Y. The genomes of bymoviruses and macluraviruses have smaller sets of the enzyme motifs as some of their N-terminal motifs are missing; Pfam (https://pfam.xfam.org/ (accessed July 2019)) found only the C-terminal four motifs of the polyprotein in the complete polyproteins of barley yellow mosaic bymovirus, cardamom mosaic macluravirus, sweet potato mild mottle ipomovirus, and wheat streak mosaic poaceaevirus. A Pfam analysis of celery latent celavirus (CLV) found only the helicase C and RdRp genes found in other potyvirids. Thus, the RdRp or the helicase proteins are probably most appropriate for inferring the phylogeny of the potyvirids.

The phylogeny of RdRp genes of all named potyvirids shows that they form at least eleven genera, of which by far the largest is the potyviruses (Figure 2). The RdRp of CLV [32,33,34] is the sister of the RdRps of all other potyvirids, but those of the bymoviruses and macluraviruses are closest to that of bufivirus UC1 and the astroviruses as they are on the shortest branches. The virions and genome of CLV have all the features of a potyvirid, not a bufivirus. Its virions are flexuous filaments around 900 nm in length and contain a single genome of 11,519 nts [34]. This confirms that their structure is likely to be closely similar to those of WMV as, using the known parameters of WMV virions, the 11,519 nts of the CLV genome will assemble with 2.304 CP subunits and form a helix 903–916 nm long [20]. The similarities and differences between CLV and other potyvirids may indicate the properties of their shared ancestor. CLV was first reported from Europe. It is sap-transmitted to several dicotyledonous plants (dicots) from several different families, but there is no record of tests of monocotyledonous plants (monocots) as hosts, and like many potyvirids, it is readily seed-borne in two plant species. CLV was not transmitted by five species of aphids. Its genome has one major open reading frame (ORF), and a minor overlapping ORF, P3N-PIPO, in the -1 frame. It has some, but not all, of the motifs found in potyvirids, but not those associated with aphid-transmission in potyvirus genomes. CLV’s one unique feature is a signal peptide at the 5′ terminus of its genome.

In summary, although the potyviruses and astroviruses share an RdRp ancestor [24], none of their other genes are related, and no more potyvirus-like or astrovirus-like intermediate ancestors are known at present. Similar conclusions can be drawn about their shared helicase and CP genes. Nothing is known at present of the origins of the other potyvirus proteins, although their diversity and likely relationships suggest that they, and especially the P1 protein, have helped generate the extraordinary diversity of the potyvirids [35]. Those unique to potyviruses may have arisen de novo, or by overprinting [36,37,38], or may have come from other organisms of which the genes have not been sequenced yet. The only safe conclusion is that the potyvirids are polyphyletic in origin.

## 3. The Potyvirus: Rymovirus Divergence

The potyviruses and rymoviruses are sister taxa (Figure 2); they diverged from a common potyvirid ancestor that, judging from their phylogenetic distances, probably had blackberry virus Y and reed chlorotic stripe virus as successive sister viruses, although the genomic sequences of both of these viruses differ significantly from those of rymoviruses and potyviruses. Although most of the differences in the genomic sequences of potyviruses and rymoviruses, or the proteins they encode, are small, they are responsible for their phenotypic differences, including their transmission by different vector types, aphids and mites. Govier and Kassanis [39,40] first reported that, for transmission by aphids, potyviruses required virions and a “helper component” present in infected plants. This was shown to be the protein now called HC-Pro, which was subsequently found to have several additional functions (reviewed by [15]), including the ability to suppress RNA silencing. Potyvirus proteins have at least three motifs associated with aphid transmission [15,41] including the DAG- motif at the N-terminus of the CP, and the -KITC- and -PTK- motifs of the HC-Pro protein; although the first two of these are not found in the homologous sites of rymovirus polyproteins, the last is. When the HC-Pro gene of mite-transmitted wheat streak mosaic tritimovirus (WSMV) was replaced by that of aphid-transmitted turnip mosaic potyvirus (TuMV), it was no longer transmitted by its mite vector [42], but there is no report of similar experiments with rymoviruses or potyviruses.

Gibbs and Ohshima [43] suggested that the divergence giving rise to the proto-potyvirus and proto-rymovirus is likely to have occurred in an infected Eurasian monocot. This conclusion was based on the fact that the primary hosts (i.e., the host from which they were first isolated) of all three rymoviruses are all Eurasian monocots; *Agropyron* (Eurasian; https://en.wikipedia.org/wiki/Agropyron (accessed 30 October 2019)), *Hordeum* (Eurasian, African, Americas; https://en.wikipedia.org/wiki/Hordeum (accessed 1 December 2019)), and *Lolium* (Europe, Asia, N. Africa; https://en.wikipedia.org/wiki/Lolium (accessed 18 December 2019)), and many of the basal potyviruses are too (Figure 3a,b). The list of potyvirus hosts closest to the rymoviruses depends on whether ORF or polyprotein sequences are used for estimating their patristic distances from the rymoviruses, but 16 of the nearest 20 are shared. Of these, 11 are from Eurasian plants (seven monocots and four dicots), and single hosts are from the Americas, Australia, Madagascar, and South Africa, and one is cosmopolitan. The possible dates of that divergence and of some of these invasions of other continents are discussed below.

## 4. Potyvirus Diversity

### 4.1. Phylogenetics

The evolution of potyviruses has been studied by using the two strategies widely used for investigating evolutionary rates and processes. Firstly, phylogenetics [44], which is based on the premise that organisms evolve by mutation and selection, so that the resulting successive divergences can be represented as a tree [45], and revealed computationally and quantified by comparing their properties, especially, nowadays, those of their gene sequences and the proteins they encode. Secondly, methods of population genetics can be used (see below in Section 4.2).

The ML phylogeny (Figure 3a,b) shows that most, but not all, of the lineages proposed in earlier published phylogenies (e.g., Gibbs and Ohshima [43]), are confirmed. The relationships between the different potyviruses in the phylogeny are also closely similar to those in a published neighbor joining (NJ) tree phylogeny [13], despite differing in relative branch lengths. Several of the virus lineages in the ML phylogeny are evident because they are subtended by long branches (Lineages 4, 5, 7, and 9); ML trees often define clusters more clearly than NJ trees, as their basal branches are relatively longer than their tip branches. Their long branches represent periods of the past that have only one known survivor, namely the lineage progenitors. Some lineages are additionally defined by the relationships of their primary hosts, namely the hosts from which they were first isolated, when the hosts are grouped at the “Order” level of the angiosperms (https://en.wikipedia.org/wiki/APG_IV_system (accessed 16 September 2019). Most viruses of Lineages 1 and 9 were first isolated from lilioids, most of Lineage 2 from commelinids, Lineage 4 and 7 from rosids, and Lineage 5 and 8 from asterids, whereas the hosts of some large lineages were from two plant clades, such as Lineage 3 which was from monocots (both alismatid and lilioid) or rosids, but rarely asterids, and Lineage 6, which was from a mixture of lilioids or asterids, but not rosids. These specificities are interesting as early attempts “to find some logic in the confusing issues of experimental host ranges of plant viruses” [46] were not resolved by more data of the experimental hosts [47,48,49]. However, the potyvirus lineages of primary hosts shown in Figure 2 and Figure 3a,b, and the even more exact correlations shown by the tobamoviruses and their primary hosts [50], indicate that there are phylogenetically influenced components of both virus and host that control their biochemical compatibility. It might be that the greater molecular repertoire of monocots, asterids, and rosids, which have resulted from repeated genome duplication [51], allow viruses to form functioning relationships with them. It is noteworthy that few potyviruses have been reported from caryophyllids and very few from basal angiosperms, such as the ranunculids, although hibbertia virus Y is from a Gunnerid, and catharanthus mosaic virus was isolated from *Welwitschia mirabilis* (a gymnosperm; [52]). Two poty-like metagenomes have recently been reported from unexpected hosts (snails and a “dipteran mix”) and are discussed below. Overall, the primary ‘host preferences’ of potyviruses among green plants is strikingly similar to the host preferences of their principal vectors, which are heteroecious aphids, that alternate between woody and herbaceous hosts, which are mostly rosids, asterids, and commelinids (grasses), but not caryophyllids [53].

Genetic recombination is common in potyvirus populations [54] and in potyvirus experiments [55]. The populations of four potyviruses discussed in Section 4.2 below, had 35% to 64% recombinants, and most reports conclude that recombination is an important factor driving the evolution of potyvirus populations [56]. However, there have been few reports that potyvirus species are recombinants involving other described viruses. Therefore, RDP version 4.95 was used to analyze the 152 ORF sequences used for Figure 3, but only five recombinants were found. Hubei poty-like virus (Lineage 2; NC_032912) was the recombinant with greatest statistical support, but is probably an in silico contaminant, which will be discussed below (see Metagenomes, Section 6.10). The analysis confirmed that WMV (Lineage 3; NC_006262) is a recombinant with soybean mosaic virus (SbMV; NC_002634; Lineage 3) as a major parent with a minor 5′-terminal region closest to bean common mosaic virus (BCMV; NC_003397; nts 1-c.770), as previously reported by Desbiez and Lecoq [57], and that Sudan watermelon mosaic virus (Lineage 4; NC035459) is also a recombinant with parents from Lineage 4, namely zucchini shoestring virus (NC_043172) and a minor 5′-terminal region close to wild melon vein banding virus (NC_035458; nts 1-c.550), as reported by Desbiez et al. [58]. Two other recombinants are novel but less certain. The ORF of calla lily latent virus (CLLV; EF105298) is mostly related to that of SbMV, but has a 5′-terminal region distantly related (74.9% ID) to that of konjac mosaic virus (KMV; NC_007913; nts 1-c.570), and likewise most of the ORF of vanilla mosaic virus (VMV) is related to that of BCMV but it has a 5′-terminal region (nts 1-c.665) that is also distantly related (61.8% ID) to that of KMV; these links were not resolved by direct nBLAST searches. These results indicate that, at most, only four of the 152 distinct potyviruses, we compared, were generated by recombination. 

The ORF phylogeny in Figure 3a,b is closely similar to a ML tree of the polyprotein sequences encoded by the ORFs, although a graph comparing their patristic distances (Figure 4) shows that there are differences of up to 15% in individual branch lengths. 

The patristic distances graph (Figure 4) also shows that the relationship between the ORF and polyprotein patristic distances is broadly linear except at the smallest axial values. This is perhaps evidence of mutational saturation, but not translational saturation; the rate changes may reflect the time-dependent bias in rate estimates of nt sequence change [59,60].

### 4.2. Population Genetics

As mentioned above in Section 4.1, the evolution of populations of organisms can be studied not only by phylogenetics, but also by using the methods of population genetics (popgen) [61,62,63]. These describe features of gene populations, using mathematical models, and compare observed features with those likely to result from sequential random changes. Popgen analyses have been used to study potyvirus populations within plant populations (e.g., Achon [64]; Li et al. [65]; Wang et al. [66]; Hajizadeh et al. [67]) and, increasingly, virus populations within individual plants (e.g., Cuevas et al. [68]; Domingo and Perales [69]; Dunham et al. [70]; Kutnjak et al. [71]; Rousseau et al. [72]; Seo et al. [73]). Such studies have shown that the effective populations of potyviruses are all around 10,000 [74], and the “The high potential for genetic variation in plant viruses need not necessarily result in high diversity of virus populations. There is evidence that negative selection results in virus-encoded proteins being not more variable than those of their hosts and vectors. Evidence suggests that small population diversity, and genetic stability, is the rule” [75]. We have investigated whether popgen analyses of the genome sequences now available for five potyviruses with contrasting biologies can reveal features that correlate with their contrasting biological differences. Note that we use the terms phylogroups (PVYs), strains (PPVs), and groups (TuMV) somewhat interchangeably, as these are the terms used in quoted publications. The viruses we examined are as follows:

#### 4.2.1. Potato Virus Y

PVY, which originated in South America [29] from a lineage of potyviruses (Figure 3b; Lineage 5) found mostly in solanaceous weeds and crops, and asteroid species in the Americas [76,77]. Potatoes (*Solanum tuberosum*) were domesticated in South America and first taken to Europe in the 16th century. They became a major international vegetatively propagated crop after the mid-19th century. PVY, which is mostly spread by locally migrating aphids and planting infected seed tubers, is now common in the potato crops of the world, where recombinant PVYs have become particularly damaging.

#### 4.2.2. Turnip Mosaic Virus

TuMV, which is the only dicotyledon-infecting member of a lineage of potyviruses of lilioid monocotyledons (Figure 3b). It diverged about a millennium ago from a virus of European orchids, currently referred to as the OM TuMV outgroup [78]. TuMV probably emerged to infect its crop and weed hosts during the development of agriculture in Eurasia in recent centuries. It is common in crops of several annual brassicas (canola, turnips, radish, etc.), that are grown from seed. It is also common in various perennial domesticated and weed *Brassicaceae* [79]. TuMV has been reported as being seed-borne in some hosts [80,81], but this has not been confirmed, and it is probably mostly spread by aphids migrating within and between populations of its perennial and annual hosts.

#### 4.2.3. Plum Pox Virus

Plum pox virus (PPV), which is a potyvirus of long-lived woody domesticated and wild *Prunus* fruit trees and shrubs. Its host populations are therefore much smaller and longer lived than the host populations of PVY and TuMV. PPV is usually considered to be a single species (Figure 3b), even though it has seven distinct strains. Its center of emergence was probably Eurasia [31]. It is spread by aphids within a wide range of wild and domesticated *Prunus* species and non-*Prunus* weeds. PPV is also graft transmitted, and most of its crop hosts have been propagated by grafting for the last three millennia. 

#### 4.2.4. Sweet Potato Potyviruses

The SwPVs, or sweet potato potyviruses (Figure 3b; Lineage 8), which have been isolated from sweet potato (*Ipomoea batatas*) crops in all continents, except Antarctica. The SwPVs are usually considered to be five species [28], but as all have only been isolated from sweet potato, we included them as it is possible that they are a single diverse mega-species, and it is of interest to check whether or not population genetics methods agree.

#### 4.2.5. Hardenbergia Mosaic Virus

Hardenbergia mosaic virus (HarMV) is a potyvirus found in the wild only in southwestern Australia where it is widespread in the perennial native legume *Hardenbergia comptoniana* [82]. It is a member of the BCMV lineage (Figure 3a; Lineage 3), which mostly radiated in South-East Asia [83], and may have been carried to Australia by Austronesian boat people. This lineage probably arrived in Australia long before Europeans arrived in 1788, colonized the continent, and subsequently developed large-scale agriculture. Thus, HarMV is a virus with smaller known host and vector populations than the other four potyviruses considered here. Its biology is discussed in more detail below.

A summary of the results of the popgen analyses of the non-recombinant (n-rec) ORF sequences of these five viruses is shown in Figure 5, which includes an outline ML phylogeny of the ORF sequences of the five viruses, and parameters from a graph comparing their ORF and polyprotein patristic distances, comparable to Figure 4. The pairwise nt diversity for each virus (π) is, as expected, related to the maximum pairwise ORF patristic distance (max ORF-dist), and the horizontal width of the corresponding collapsed cluster in the tree. The ω values (dN/dS) for four of the viruses are in the range of 0.065 (PVY) to 0.086 (PPV) confirming that all are under strong negative selection; however, the value of 0.260 for the SwPVs indicate that these viruses are under less stringent negative selection and/or are not evolving as a single population, i.e., providing evidence that they are separate species [84]. The slope (b) of the linear regression in a patristic graph comparing ORF and polyprotein trees (Figure 4) correlates with ω. The slope varied from 0.243 to 0.450 for the five individual viruses (Figure 4). The slope was even greater for between-virus comparisons, and, for the 152 potyviruses represented in Figure 3a,b, it increased from c. 0.6 at the smallest axis values to c. 1.3 (Figure 4). 

The popgen comparisons of the different phylogroups/groups/strains of PVY, TuMV, SwPVs, and HarMV isolates are given in Appendix A and summarized in Figure 6. The data for PPV are from Hajizadeh et al. [31]. All the groupings are monophyletic; the BRs group of TuMV is normally subdivided into a basal paraphyletic “basal-BR” group with a subgroup called “Asian-BR”. It can be seen that around three-quarters of the groups are represented by 10 or more sequences. The diversity of each group (π) correlates with their collapsed cluster sizes with PPV groups having the smallest diversities and HarMV group/species having the largest. The ω ratio estimates for individual genes (Figure 7) are considered to assess the strength of selection against translational change in the encoded protein. These fall into three groups; the proteins encoded by the n-rec genomes that are most strongly selected (smallest ω) are the HC-Pro, CI, Nia, and NIb proteins (ω = 0.047–0.055), an intermediate group is of the P1, P3, VPg, and CP proteins (ω = 0.101–0.229), and PIPO is least conserved (ω = 0.526), whereas the rec genomes give ω values that are around 10% greater, and in the same groupings except that VPg is one of the most strongly conserved proteins.

Comparisons of the popgen results for the five viruses are instructive. There are five major phylogroups in the world PVY population (Figure 6), three of them (O, C, and N) are mostly n-rec, and two populations, R1 and R2, are based on single and double OxN recombinants. Commonest isolates from most potato-growing areas, and the most frequently sequenced, are O, R1, and R2 isolates. ORF sequences of the C phylogroup, which mostly have been isolated from a range of non-potato solanaceous hosts, are much more diverse (Appendix A; π = 0.113) than those of the O and N phylogroups (π = 0.027 and 0.043), which were mostly isolated from potatoes, and this diversity is also shown in the number of segregating sites (S), mutations within segregating sites (η), and the average number of differences between sequences from the same populations (k). The ω ratios for the individual ORFs (Appendix A), and for each of their genes separately (Appendix A), are less than one, indicating that their genomes are under dominant negative (purifying) selection; those of the n-rec isolates under stronger selection than those of the rec isolates (average ω 0.147 compared with 0.208), and with the PIPO gene (Figure 7; yellow bars) under least selection. The phylogroups are genetically distinct and there is no evidence of gene flow between them (Appendix A; Ks, Z, Snn, and Fst tests), confirming that after the primary recombination events which established the R1 and R2 lineages there was no significant gene flow between them. The Tajima’s D test (Appendix A) gave a significant negative value for the O, R1, and R2 populations, confirming that they had recently expanded after a population bottleneck, and this reflects the adoption of potatoes as a major international crop after the mid-19th century with the O phylogroup and R1 and R2 necrogenic recombinants, but not the N and C phylogroups, as major pathogens of it. The major expansion evident in the Tajima’s D scores was confirmed in popgen analyses of most of the individual PVY proteins (Appendix A), especially the HC-Pro, P3, CI, and VPg proteins. The evidence of less-stringent negative selection of the recombinant isolates (R1 and R2 phylogroups) is shown mostly in their P1 and PIPO genes.

TuMV, which is common in cultivated and weed brassicas worldwide, has four major groups: world-B, Iranian, BRs (basal-BR and Asian-BR), and basal-B. It also has the orchid outgroup found only in Europe, as mentioned above in Section 4.2.2, TuMV-OM [78], and although currently included with TuMV, this is clearly distinct according to its genetic differentiation coefficient (Appendix A; mean Fst 0.81; [83,84]. It was therefore excluded from Figure 6. Most diverse, and basal to the brassicas infecting isolates of TuMV proper, are the basal-B and Iranian populations, and although Tajima’s D test gave negative values with both the world-B and BRs (basal-BR and Asian-BR) groups, neither was statistically significant. The TuMV population is more genetically diverse than the PVY population (Figure 5), but all TuMV genes, especially PIPO, are under stronger negative selection than those of PVY (Figure 7). These clear popgen differences between the PVY and TuMV populations probably reflect biological differences that have affected translational selection; the PVY population has recently expanded its range rapidly, mostly by trade, and into monoculture crops, whereas the TuMV population has diverged as vector aphids have moved it between and within annual crops and perennial weeds, and this greater diversity of hosts may produce greater negative selection.

The PPV population has a smaller mean nt diversity than the PVY and TuMV populations; π for PPV is 0.121 (Figure 5) and a mean of 0.020 for its strains (Figure 6), but it has similar ω values. These differences probably result from its smaller population living in a long-lived woody host and transmission by grafting. Four of its major strain populations (D, M, CR, and C) have negative Tajima’s Ds, but only that of the D strain is statistically significant, and this is the strain that spread during the 20th century across Europe and from there to the Americas and East Asia, whereas the others are mostly confined to Eurasia, which is probably the “center of divergence” of PPV. The ω values for the genes of different PPV strains (Figure 7), especially those with larger sample sizes, are dominated by large values for PIPO, but it can be seen that the conserved genes of PPV (HC-Pro, CI, NIa, Nib, and CP) are, like those of PVY, less conserved than those of TuMV.

The popgen results for the SwPVs indicate that they are a lineage of separate viruses found in a single host species, not a single megaspecies. The mean nt diversity (Figure 5) of the SwPVs (π = 0.258) is greater than that of the other four viruses (π = 0.064–0.139), and the diversities of its component species range from 0.020 to 0.057 (Figure 6). They are also distinguished as species by the genetic differentiation coefficient (Fst > 0.872) [85,86]. The n-rec sequences of four of the SwPV populations are found worldwide, but that of sweet potato latent virus (SPLV) is not (see below), and all gave negative results in the Tajima’s D test, but only that of SPVG was statistically significant; SPLV was excluded from these analyses as only three ORF sequences of it were available. Of interest too, is that all the SwPVs have a PIPO gene, like other potyviruses, but, in addition, all, except SPLV, have a second overlapping gene, PISPO, and the ω values (Appendix A) indicate that the PISPO genes of sweet potato feathery mottle virus (SPFMV) and sweet potato virus G (SPVG) are under positive selection (ω = 1.162 and 1.336) as is also the PIPO gene of sweet potato virus C (SPVC) (ω = 1.253).

The HarMV population was represented by a single population of 10 ORF sequences. It is the only ‘wild plant’ potyvirus which we examined. The results are remarkably similar to those of the other four viruses we checked. The HarMV sequences have a nt diversity (π) similar to that of TuMV and a ω value (0.078) similar to those of PVY and PPV but less than that of the SwPVs (Figure 6, Figure 7 and Appendix A). The ω values for individual HarMV genes (Figure 7) have a pattern within those of the other viruses, and with no evidence from Tajima’s D tests of a major recent population expansion. The HarMV population was more variable than that of Mediterranean ruda potyvirus (MeRV), which has a similar ecology [87].

In summary, the results of the popgen analyses complement and confirm inferences obtained from phylogenetic analyses of the same ORF sequences (see above in Section 4.1). They also suggest that various popgen estimates, such as ω, Fst, and b, might be used as indicators of whether a population of similar viruses is one or more species. This could provide a more theory-based way of defining which viruses form natural potyvirus species, and hence could replace the present unsatisfactory method based on arbitrary maximum %ID values (<76% nt identity and <82% amino acid identity; [88]). Estimates of various popgen parameters indicate that the SwPVs are probably separate species, as too are the brassica-infecting and orchid-infecting TuMVs. The possibility of using popgen parameters to define species was tested further by checking whether the fast-evolving isolates of the C phylogroup of PVY [29] were a separate species. These 28 extra sequences were added and the popgen factors recalculated, but only marginally changed the popgen parameters, indicating that PVY is still a single species. Further tests were also made using the ORF sequences of 16 isolates of narcissus yellow stripe-like (NYSLV) viruses, which fall into five distinct clusters, although biologically indistinguishable. Ohshima et al. [89] reported that it was uncertain whether the NYSLVs should be considered one species or more, as their ORF sequence diversity is at the %ID limit set by the ICTV for delineating potyvirus species. The ω values for the four largest clusters, each of three sequences, had a mean of 0.055 (range 0.054 - 0.058), but as the clusters were progressively clustered, as in their ML tree [89] (Figure 2) the value of ω increased to 0.117 > 0.129 > 0.146 > 0.154 and finally 0.157 with the addition of the ORF of wild onion symptomless virus. These comparisons suggest that the NYSLVs are five independently evolving viruses, and that a calculated ω value greater than 0.1 is a useful indication that a population of potyviruses consists of more than one species.

## 5. Potyviruses in Space and Time

Attempts to add the dimensions of time and space to phylogenies of organisms, especially viruses, are currently a very active area of research [9]. Although virus genomes can be recovered from preserved plant specimens, the maximum age of such recoveries is only around 1000 years [90]. However, it is also possible to determine the age of virus populations that are evolving at a measurable rate by comparing the gene sequences of isolates collected “heterochronously” over the longest possible time period and analyze them by using Bayesian Monte Carlo coalescent methods in the BEAST packages [91], or by regression methods such as “Least Squares Dating” [92]. Bayesian methods also now extend to assessing virus migration pathways, using SPREAD [93], etc. These methods have been used for elegant analyses of the spread of maize streak mastrevirus [94], tomato yellow leaf curl begomovirus [95,96], and of rice yellow mottle sobemovirus [97,98], all of which, importantly, corroborate their conclusions using historical records of the spread of the diseases they cause. Studies on migration pathways for potyviruses are limited to TuMV [99,100].

Studies of timescales have been made with several potyviruses. CP genes were used in early estimates of the “time to most recent common ancestor” (TMRCA) of the zucchini yellow mosaic virus (ZYMV) population [101], the papaya ringspot virus (PRSV) population [102], and of the initial radiation of all potyviruses [103]. These studies gave compatible dates of 408 years before present (YBP), 2250 YBP, and 6600 YBP, respectively. CP genes were also used to obtain a recent estimate [104] of 129–169 YBP as the age of the present narcissus late season yellows virus population. However, most dating studies of potyviruses have been of PVY and TuMV populations (Table 1) and are based on gene sequences varying from full length ORFs of more than 9000 nts to the VPg gene of only 564 nts [29,30,100,105,106]. The results show an interesting and unexpected positive correlation between the length of the sequence and its estimated TMRCA (Table 1); the longer the sequence analyzed, the older the apparent age of the common ancestor.

External corroborative evidence for the PVY and TuMV dating studies is imprecise. TuMV studies [99] estimated that the virus arrived in Australia and New Zealand from Europe, at dates that were consistent with records of its first appearance in those countries, and the timescale obtained from the complete ORFs of PVY [29] dates major divergences of its two main lineages to the mid-19th century, and hence they coincide with a major European effort to breed potato late blight (*Phytophthora infestans*)-resistant potatoes and a significant increase in European potato cropping.

The estimated TMRCA of the population of one potyvirus can be extrapolated to those of other potyviruses, if their gene sequences align, and the alignment is used to calculate and compare them in a single ML tree; Fuentes et al. [29] in their Figure 4 showed that dates obtained from a ML tree and a Bayesian maximum clade credibility tree are linearly related. This is the basis of the “sub-tree comparison” method first used by Mohammadi et al. [107], who showed that the TMRCA of the known world beet mosaic virus (BtMV) population is compatible with a 19th-century emergence of BtMV from wild beet rather than an ancient infection of chard or leaf beet. It was also used to show that the TMRCA of PPV was probably around 820 (range 865–775) BCE [31], which is compatible with the invention of fruit tree grafting at the beginning of the first millennium BCE [108]. When aligned representative sequences of TuMV, PVY, PPV, and the basal groups of the potyvirus tree were used to calculate an ML tree (Appendix A), the divergence of the potyviruses and rymoviruses was dated, using the TuMV TMRCA (Table 1), as 14,206 YBP, and using the PVY TMRCA, as 30,192 YBP. The later date is interesting as it suggests a possible route by which the PVY lineage invaded the Americas (see below, Section 6.5).

## 6. Evolutionary Vignettes from Potyvirus Studies

### 6.1. Super-Adapters

The conclusion that the potyviruses originated in Eurasia only a few tens of thousands of years ago is important, as it helps define the conditions that have allowed potyviruses to migrate and become the important and speciose genus it is nowadays. The hallmark property of potyviruses seems to be their ability to infect and maintain populations in a wide range of plants and to switch hosts often, but be constrained to particular clades of eudicotyledons. The divergence of potyviruses has occurred during the period when humans and their activities have come, increasingly, to dominate the world, and the ability of potyviruses to exploit those conditions have allowed them to become numerous and often damaging. Although humans originated in Africa around 300,000 YBP, they did not migrate beyond Africa’s immediate environs until c. 80,000 YBP, then migrating to Asia and Australia, and eventually the Americas. Trade is an ancient human activity (https://en.wikipedia.org/wiki/Trade#Prehistory (accessed Dec 6 2019)), and seed-borne pathogens, such as potyviruses, are likely to have been carried along the ‘Spice Trade’ routes of the Indian Ocean, the Silk Road of Central Asia for many centuries. However, arguably the most important factor contributing to the success of the potyviruses and geminiviruses has been worldwide marine trade, which started with the exploration of the Americas by Columbus in the 15th century and resulted in the major exchange of plants and animals between the Americas and other regions of the world known as the Columbian Exchange (https://en.wikipedia.org/wiki/Columbian_exchange (accessed 6 December 2019)). Most recently, travel and trade by air has become important in the worldwide movement of viruses and their hosts despite an increased awareness of the importance of quarantine.

### 6.2. Australian Potyviruses

A large number of potyviruses have been found on the isolated continent of Australia, with many of them being found only there. They provide insights into the timing of potyvirus migration, as Australia was biologically isolated for many millions of years (https://en.wikipedia.org/wiki/Wallace_Line (accessed 16 December 2019)). Australia was first colonized by the Aborigines at least 40,000–65,000 years ago [109] (https://en.wikipedia.org/wiki/History_of_Indigenous_Australians (accessed 10 December 2019)). It was linked to Papua New Guinea by a land bridge during the last Ice Age, forming the continent of Sahul, until this bridge was flooded 6500–8000 years ago [110]. The Austronesian voyagers from Asia visited the north coast of Sahul during 3000–4000 YBP, but not until the 17th century did the Macassans collect trepang or bêche-de-mer and camp on the beaches of Northern Australia. Europeans explored the entire coastline of the continent around the same time and established permanent colonies in the 18th century. Regular trade between Australia and other parts of the world started about two centuries ago and has increased ever since. The spreading of potyviruses depends on the ‘nonpersistent’ transmission by migrating aphids, especially those of the dioecious *Aphidini*, which, although much more ancient [111], have also been favored by the development of agriculture. Australia had, until recently, a very small population of potential aphid vectors of potyviruses (i.e., *Aphidoideae*), with no more than 20 endemic species; however, there are now around 200 other species of the 4700 mostly recorded in the temperate Holarctic regions of the world [112,113,114,115].

Potyviruses constitute the largest virus group with representatives known from only Australia, with most being represented only by single partial nt sequences in GenBank. They fall cleanly into one or other of two groups. There are those commonly found in crops, and also found in crops in other regions of the world, and to which they are closely related. These potyviruses most probably entered Australia recently in trade. There is also a large number of potyviruses found in native or introduced weed species, but these are more distantly related to potyviruses from other regions of the world. Most of the latter are members of the BCMV group (Group 3, Figure 3a), whereas the recent migrants are from various groups. The Australian BCMV group potyviruses form two clusters. One consists of ceratobium virus Y (CerVY), dianella chlorotic mottle virus (DiCMV), euphorbia ringspot virus (ERV), glycine virus Y (GVY), kennedya virus Y (KVY), passiflora foetida virus Y (PfoVY), passiflora virus Y (PaVY), pleione virus Y (PleVY), pterostylis virus Y (PtVY), rhopalanthe virus Y (RhoVY) and sarcohilus virus Y (SarVY), and the other of clitoria chlorosis virus (ClCV), clitoria virus Y (CliVY), diuris virus Y (DiVY), eustrephus virus Y (EustVY), HarMV, hibbertia virus Y (HibVY), passiflora mosaic virus (PaMV), passionfruit woodiness virus (PWV), and siratro viruses 1 and 2 (Sir1VY and Sir2VY) [11,82,116,117,118,119,120,121,122]. This suggests that there were two near-simultaneous incursions into Australia of viruses of the BCMV group, and the fact that PaVY is also found in New Guinea [120] supports the suggestion [83] that the early BCMV group viruses were brought from Asia to the north coast of Sahul by Austronesian speakers who were not only explorers but also colonizers; they traveled in outrigger canoes with various live domesticated plants and animals so that they could establish viable communities when they found habitable islands.

Recent studies using high throughput sequencing of plants belonging to the Southwest Australian native flora detected three additional potyviruses of orchids: blue squill virus A (BSVA), donkey orchid virus A (DOVA), and caladenia virus A (CalVA) [123,124,125]. BSVA group with HarMV and PWV within the BCMV potyvirus group [125], whereas CalVA and DOVA do not [103,125], and their date and mode of entry to Australia is unknown.

### 6.3. Papaya Ringspot Virus

The spread of PRSV around the world is also an informative story. It was first recorded in Hawaii, by Jensen [126], and has been reported since then from most of the world’s tropical and subtropical regions. PRSV is a typical potyvirus, as it is transmitted non-persistently by several aphid species [127], and also occasionally in seeds of papaya (*Carica papaya*) [128] and *Robinia pseudoacacia* [129]. In 1984, it was shown to be closely related to one of the potyviruses common in wild and cultivated cucurbit crops worldwide, and it was called watermelon mosaic virus 1 at that time [127]. These viruses are now called the PRSV-P (papaya) and PRSV-W (watermelon) biotypes of a single species; PRSV-W infects cucurbits, but not papaya, whereas PRSV-P infects both papaya and cucurbits. Cucurbitaceous weeds are natural reservoirs of infection for crops of both PRSV biotypes [130,131]. PRSV is closely related to several other viruses of cucurbits in the Middle East and Africa (Group 4, Figure 3a; [58,102,103,118]), one of which is recorded to infect papaya in the Congo, causing serious disease including ringspots [132]. PRSV probably arose in South Asia and spread from there to the remainder of the world, including the Americas [58]. An isolated outbreak of papaya ringspot disease in Australia was particularly informative as phylogenetic studies of the CP gene sequences of PRSV from the outbreak [133,134] grouped with sequences from the long established local PRSV-W population, rather than others from overseas. Thus, the local PRSV-W population was the likely source of the PRSV-P outbreak. Olarte-Casillo et al. [102] questioned this conclusion when they used Mesquite 4.7 [135] to make an “ancestral reconstruction” of the PRSV population, using all known CP sequences, and concluded that “ancestral state could be either of the two biotypes”. Ancestral reconstruction assumes, however, that the data used are a representative sample of the world PRSV population, and c. 70% of the sequences they analyzed were of PRSV-P biotype, whereas surveys of 22 Pacific Islands [136], where papaya and various cucurbits are commonly grown for food, found PRSV-W on 17 different islands, but PRSV-P on only four. More recently, Maina et al. [137] found evidence that only one PRSV-W introduction, or multiple introductions of very similar isolates, has occurred to Australia since agriculture commenced following the arrival of European colonists. Thus, PRSV-W genomic sequences from the entire northern coastline of Australia most resembled those from Papua New Guinea (PNG), but differed from those from East Timor (ET) and elsewhere.

It seems that the PRSV-W to PRSV-P conversion occurs very infrequently. It was suggested [138] that this host shift is controlled by the VPg-NIa-NIb region of the genomes of experimentally constructed PRSV-W x PRSV-P recombinants. Comparisons showed that simultaneous changes in three amino acids in the NIa were most likely involved: an Ala (W) to Val (P), an Asp (W) to Lys (P), and a Val (W) to Ile (P), all involving one or more first or second codon position mutations, which is an unlikely event.

### 6.4. The Columbian Exchange

It is clear that the host ranges and distribution of potyviruses have been greatly affected by the Columbian Exchange, mentioned above, in Section 6.1. For example, PRSV, as described above, in Section 6.3, is part of a lineage of Old World cucurbit viruses. It was reported by Olarte-Castillo et al. [102] to have a TMRCA of 2250 (95%CI 9800-250) YBP, which is congruent with the report of Desbiez et al. [58] that the TMRCA of the PRSV cluster is 3600 YBP. However, papaya is a native of tropical Southern Mexico and Central America and was confined to that area until about 1550 CE (470 YBP), when Spaniards carried seeds to the Philippines, and from there to Malacca, India, and to Naples in 1626 CE, so that now papaya is commonly grown in all tropical regions of the world [139]. These facts indicate that PRSV-W is the pre-Columbian version of this virus of cucurbits, and, only recently, has papaya been spread around the world by mankind and, occasionally, where they have met “papaya ringspot disease” has been generated.

There are several other examples of “new encounter” potyvirus diseases involving a crop species from one region of the world that has been taken for the first time by mankind into the territory of a potyvirus, resulting in disease, often severe [140]. Sweet potato, for example, was domesticated in Central America [141]. It probably spread by natural means to Oceania, where it was found and exploited by the Austronesian explorers [141]. Iberian galleons took sweet potatoes and other New World crops first to the Philippines and then to China and Okinawa by the early 17th century, and later to Korea. Sweet potato arrived in Europe across the Atlantic and was recorded in England in 1604 (https://en.wikipedia.org/wiki/Sweet_potato (accessed on 10 December 2019)). There is evidence that the five related potyviruses (Lineage 8, Figure 3b) found in sweet potato throughout the world may have originated outside the Americas, long before sweet potato encountered them. SPFMV, SPVC, SPVG, and sweet potato virus 2 (SPV2) now have a worldwide distribution, judging from GenBank records, which reveal that the sequenced isolates came from Africa, the Americas, Asia, Europe, and Oceania (SPFMV, 34 countries/ 443 records; SPVC, 16/69; SPVG 19/150 and SPV2 9/22). By contrast, SPLV, which is the basal sister to the others, has 92 records, but only from China, South Korea, Taiwan, and Tibet, and all the most closely related viruses (PPV, hyacinth mosaic virus, and asparagus virus 1) are Asian or European, suggesting that the center of divergence of Group 8 is Asia, not the Americas.

### 6.5. The Major Lineages: BCMV and PVY

The BCMV and PVY lineages are the largest in the potyvirus phylogeny (Figure 3a,b; Groups 3 and 5). Both are clearly delineated by long branches, indicating a period in their history that produced no other known survivors. The BCMV lineage mostly radiated in SE Asia and Australia [83], and its primary hosts are mostly rosids and monocots, and viruses of the group have caused most damage in crop species that originated in other areas of the world: common beans (*Phaseolus vulgaris*) from central America, passion fruit (*Passiflora edulis*) from South America, cowpea (*Vigna unguiculata*) from central Africa, only soybean (*Soja max*) being from SE Asia. By contrast, most of the primary hosts of the PVY lineage are asterids, and Fribourg et al. [76] noted that all but two of the 27 viruses of the PVY lineage were isolated in the Americas, 17 of them having never been found anywhere else. Thus, it is likely that the PVY lineage diversified in the Americas, although how and when it originally migrated there from the Old World are interesting questions. If the TMRCA of PVY in Table 1 is correct, then Appendix A, a ML tree of 38 representative ORF sequences, indicates that the long basal branch of the lineage covered the period 15.4–18.8 thousand YBP. This is the period human progenitors of the Amerindians are most likely to have migrated to the Americas (https://en.wikipedia.org/wiki/Settlement_of_the_Americas (accessed 17 December 2019); https://en.wikipedia.org/wiki/Indigenous_peoples_of_the_Americas (accessed 23 Dec 2019)) from Beringia, which was, at that time, at the eastern end of the mammoth steppe biome. This biome stretched from Iberia, across the north of Eurasia to Beringia (https://en.wikipedia.org/wiki/Mammoth_steppe (accessed 11 December 2019)). Around 16 thousand YBP, ice retreated enough to open one, possibly two, routes from Beringia into the Americas [142]. The mammoth steppe was sub-Arctic and dominated by willow shrubs, grasses, and herbs, including many *Artemisia* spp. [143,144], which are asterids, like many primary hosts of the PVY lineage viruses. It seems that no one has surveyed the viruses of *Artemisia* yet, and to do so might be very instructive.

### 6.6. Genetic Connectivity

This is a biosecurity term that refers to situations where genetically similar nt sequences occur among populations of the same virus obtained from infected plants growing in different countries [14]. In studies with potyvirus isolates from infected crops in Northern Australia and nearby countries ET and PNG, genetic connectivity was demonstrated in three instances. These were between one location in Northern Australia and ET for ZYMV [145], two locations in Northern Australia and ET for SPFMV [146], and, as mentioned above in Section 6.3, locations spanning the entire region of Northern Australia and PNG for PRSV [137]. Such findings indicate that important potyviruses of economically important crops are crossing the sea separating Northern Australia from nearby countries to the north. A possible explanation is that this connectivity has arisen through trade in either direction. Alternatively, it may have resulted from spread by viruliferous insect vectors blown across the sea by annual monsoonal wind currents. These findings emphasize the need for improved biosecurity measures to protect against potentially damaging international virus movements occurring by natural means in addition to those resulting from international trade in plant materials [137,145,146].

### 6.7. Seed Transmission

Several of the most important crop potyviruses have been shown to be seed-borne [16,147]; however, many other potyviruses have not. Regardless, there are many records of crop potyviruses being found that could only be logically explained by seed transmission; indeed, it is likely that potyviruses are more frequently seed-borne than published records suggest perhaps because inadequate numbers of seeds were tested [148]. There are many examples of potyviruses being spread from one part of the world to another by the international seed trade [14,16]. Moreover, sowing potyvirus-infected seeds results in multiple infection foci consisting of infected seedlings scattered at random throughout crops. This constitutes a critical source from which aphid vectors can acquire a potyvirus and spread it throughout a crop resulting in serious virus disease epidemics [149,150,151].

### 6.8. Potyvirus Emergence from Natural Ecosystems

There is evidence of agriculture being practiced by Australia’s aboriginal population well before the island continent was first colonized by Europeans in 1788. The crops they grew consisted of Australian native plants, such as wild yams and grasses grown for their storage roots and seeds, respectively [152]. Interfaces between natural vegetation and introduced crop species only arose after 1788, and, in many parts of the continent, are much more recent than that. Such interfaces are therefore well suited to studies on potyvirus emergence from native vegetation to infect introduced crop plants and weeds, and vice versa [140]. Two such studies documented the spread of HarMV from infected *H. comptoniana* plants to nearby plants of introduced lupin species [153,154]. PWV may have spread from its indigenous Australian host *Passiflora aurantia* (golden passion flower) to cause widespread infection in recently introduced crop, forage and naturalized weed legumes and *Passiflora* spp., which originated in South and Central America, or East Asia [103,121,122,140,155]. PaMV and PaVY have been found infecting the passionfruit crop and naturalized introduced weeds so presumably also spread to them from an as yet unknown native plant host. Similarly, the indigenous Australian potyviruses of *Clitoria* and siratro (*Macroptilium atropurpureum*) have so far only been isolated from naturalized introduced weeds but presumably spread from unknown native plant hosts to introduced species after European colonisation [82,118,155]. CerMV, DiVY, EustVY, GVY, HibVY, KVY, PleVY, and PtVY, have so far only been found infecting native plant species [118,155], whereas RhVY and SarVY have only been isolated from imported orchids [118].

In situations where a stable mixed plant population infected with viruses has co-evolved with native wild plants over a significant period in a given world region, populations of its isolates collected over a limited geographic range are likely to be diverse (e.g., Spetz et al. [156]). Isolates of the most studied Australian indigenous potyviruses, HarMV, PaVY and PWV, provide excellent examples of this phenomenon [82,122]. The localized natural distribution of HarMV reflects that of its Australian principal native host plant *H. comptoniana*. PWV and PaVY occur most commonly in the warmer tropical and subtropical regions of the Australian continent, where, as described above in Section 6.2, they infect a range of hosts. When ClVY, HarMV, PaVY, and PWV were inoculated to Australian native plant species, some that became infected produced surprisingly severe systemic symptoms. This observation did not support the suggestion that indigenous viruses are likely to be harmless when they infect wild plants. Instead, it indicates that, when they encounter native plant species they are not adapted to, they will likely behave in the wild just like crop viruses behave when they infect new crops [157].

### 6.9. Historical Potyvirus Specimens

Historical collections of plant viruses are proving extremely valuable in linking the pre- and post-sequencing eras of potyvirus research. This applies not only to providing sequence data for dating studies described above in Section 5, as these require comparisons between the sequences of old and recent potyvirus isolates covering the largest possible timespan, but also in revealing whether potyviruses studied in the post-sequencing era are named correctly, and if new sequences from unpublished potyvirus isolates from historical collections can be linked with those of established viruses.

Studies on historical isolates preserved since the 1970s and 1980s from the Andean region of South America and Europe provide examples. With PVY, a dating study without these historical isolates suggested the “time to most common ancestor” (TMRCA) of PVY was the 16th century [105], whereas a similar study which included sequences from European isolates from the 1980s [158] placed its TMRCA around 1000 CE [106]. When new PVY isolates from the Andean potato domestication center were included in dating analyses, along with the 1980s European sequences, the TMRCA was estimated to be around 156 CE [29]. This study concluded that, although PVY was first taken to Europe in the 16th century in potato tubers after the Spanish conquest of Peru, as mentioned above in Section 5, most of the current PVY diversity developed in the mid-19th century, when additional potato breeding lines were imported to help develop potatoes resistant to the potato late blight pandemic then occurring.

Historical samples were also valuable when the sequence of a historical arracacha virus Y (ArVY) isolate collected in Peru in 1976 was compared with other potyvirus sequences, as it had 79% nucleotide identity with a 2013 Brazilian isolate of the subsequently described arracacha mottle virus, showing the latter was actually ArVY [159]. Similarly, when a potyvirus sequence obtained from a 33-year-old mashua (*Tropaeolum tuberosum*) sample was sequenced and compared with other potyvirus sequences, it was found to be a distinct potyvirus to which the name mashua virus Y (MasVY) was given; MasVY’s relationships to three other potyviruses (Tropaeolum mosaic virus, Tropaeolum virus 1 and Tropaeolum virus 2), described previously from the same host, are unknown, as the genomes of these viruses are still not sequenced [160]. Finally it is noteworthy that two isolates of a potyvirus collected from pepino (*Solanum muricatum*) in Peru in 1976, and kept in a historical collection thereafter, have been recently shown, along with five new Peruvian pepino potyvirus isolates, to be isolates of wild potato mosaic virus, a virus previously known only from the wild potato *Solanum chancayense* [76,161].

### 6.10. Metagenomes, A New Frontier?

Recently, new methods of gene sequencing have permitted surveys of virus-like genes and genomes in environmental samples (soil, water, air, etc.) and in bulk cellular material (blended invertebrates, etc.). The resulting metagenomic sequences have greatly expanded the size of the known virosphere [24,162].

Two metagenomes appeared in the genomic sequences downloaded from GenBank, using the search term “Potyviridae”, and both were reported from China. One is of Wuhan poty-like virus 1 (KX884573), which groups with the macluraviruses (Figure 2). It was isolated from the blended remains of 12 Chinese land snails (*Mastigeulota kiangsinensis*), collected in Wuhan, and has 73% ID with 95% cover of the genome of broad-leaved dock virus A (NC_038560) from *Rumex obtusifolius* from New Zealand. It is more distantly related to yam chlorotic necrosis virus (MH341583) from *Dioscorea alata* from India. Thus, Wuhan poty-like virus is more likely to be a constituent of the snails’ last supper than a virus infecting the snails. The other metagenomic potyvirid is Hubei poty-like virus 1 (NC_032912), which was obtained from a “dipteran mix” of insects collected around Hubei. It is a recombinant with a major parent (nts 1-7630, 98% ID; 84% of the sequence) closest to an unusual isolate of sugarcane mosaic virus from *Canna* sp., in China (KY548507), and a minor parent (nts 7631-9094) closest to an unusual isolate of bean yellow mosaic virus (DQ060521; 95% ID) also isolated from “naturally planted” Canna sp., in China. This again suggests a contaminant virus, not an infection of the dipterans in the source material. Thus, both potyvirus-like metagenomes are more likely to be from contaminating plant materials than from the tissue of the animals being tested. Furthermore, Hubei poty-like virus 1 is probably an in silico recombinant, as its likely recombination site is adjacent to the region encoding the -GNNSGQP- motif, which is a conserved potyvirid motif [11,12], and both ‘parents’ detected by BLASTn had been isolated from Canna plants.

## 7. Conclusions

The major value of the esoteric studies of potyviruses described in this review is to provide a coherent evolutionary framework in time and space within which it is possible to plan how best to minimize the damage done by potyviruses to plants valued by humankind. Such a framework also contributes to the understanding of how potyvirus infections interact with plant populations within managed and natural ecosystems. Moreover, knowledge of the relationships of each virus, especially a newly emerged one, can provide valuable predictions of its likely behavior by extrapolating the known properties of near relatives and adding a timescale to those inferences greatly enhances their value and likely accuracy.

The story in our review started in 1931, with PVY, one of the first two described viruses of potato, but now, nine decades later, at least 49 more viruses of potato have been recorded [163]. The group of viruses, of which PVY is the type, is now represented in GenBank by genomic sequences of more than 150 different potyviruses and a third as many other potyvirids. Their discovery represents a considerable international effort as an analysis of the 2610 ‘country’ records (Appendix A) shows that they came from 82 different countries from all regions of the world. The pioneers of potyvirology, Redcliffe Salaman, Kenneth Smith, Fred Bawden, Basil Kassanis, et al. discovered the first of a fascinating lineage of viruses that has provided us and others with an enormous amount of interesting and valuable work with no sign that the lode will ‘run out’ anytime soon.

## Figures and Tables

**Figure 1 viruses-12-00132-f001:**
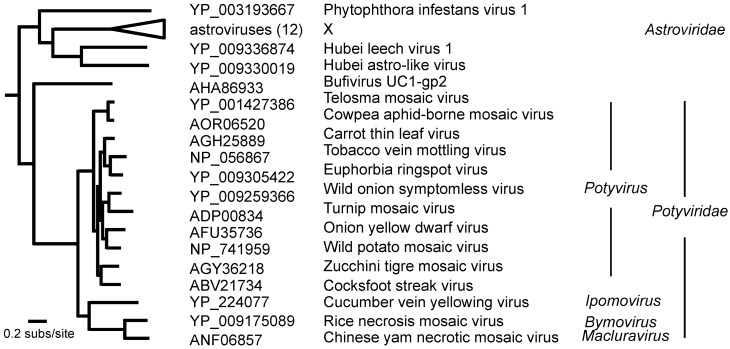
The phylogeny of the RdRps of selected potyvirids, and of the other most closely related viral RdRps (redrawn from [24]).

**Figure 2 viruses-12-00132-f002:**
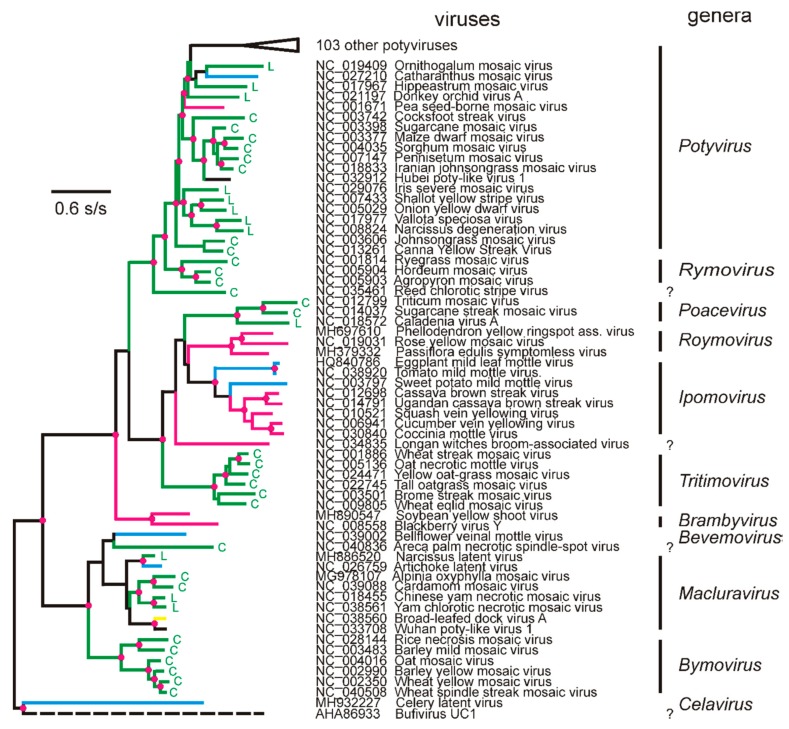
A maximum likelihood (ML) phylogeny of 166 potyvirids, calculated using the same methods as [29,30,31], from the protein sequences encoded by the RdRp regions of all genomes with GenBank Reference Sequence Accession Codes (July 2019), together with any others that had unique names. The most closely related non-potyvirid, bufivirus UC1, is included as an outlier. Branch colors indicate the major angiosperm “Order” from which each virus was isolated (i.e., its “primary host”); eudicotyledon rosid (red), asterid (blue), or caryophyllid (yellow) and monocotyledon (green) alismatid “A”, lilioid “L”, or commelinid “C”. The RdRps of potyviruses branching most closely to the rymoviruses are shown, and the phylogeny of the other 103 is collapsed. The nodes with a red disk have >0.9 SH support.

**Figure 3 viruses-12-00132-f003:**
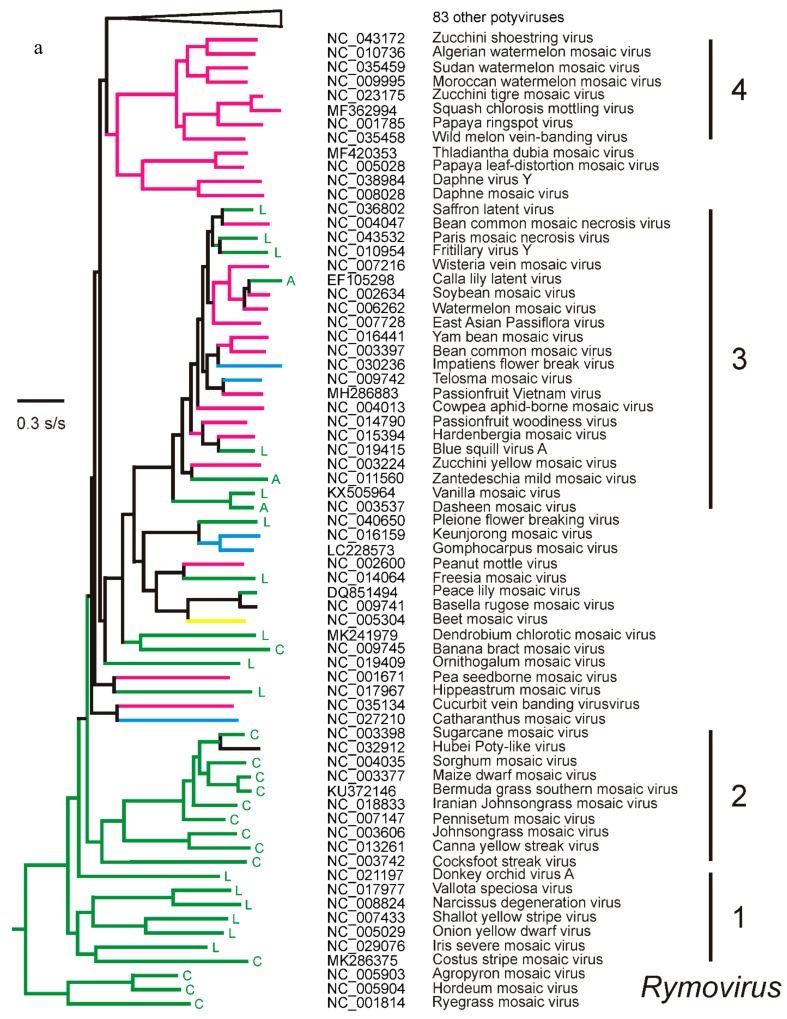
An ML phylogeny of 149 potyvirus ORF sequences with three rymovirus ORFs as outgroup using the same methods as [29,30,31]. The ORFs are from all 126 potyviruses represented in GenBank in mid-2019 by Reference Sequences, together with single representative sequences of all 23 other potyviruses in GenBank with unique names. Note that (**a**) shows the three rymoviruses and Lineages 1–4, whereas (**b**) shows Lineages 5–9. The “Order” of the primary host, namely the plant from which each was first isolated, and often given in the name of the virus, is shown by the branch color (and letter), eudicotyledonous rosid (red), asterid (blue), or caryophyllid (yellow) and monocotyledonous alismatid “A”, lilioid “L”, or commelinid “C”.

**Figure 4 viruses-12-00132-f004:**
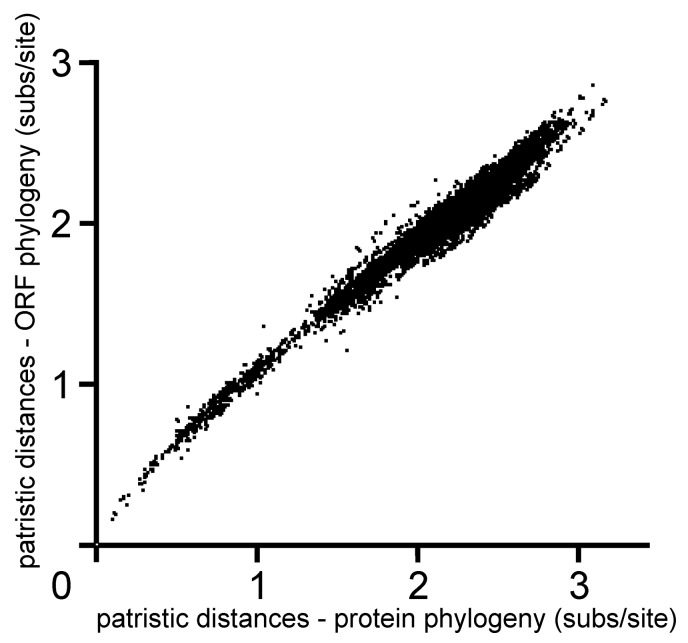
A graph comparing the patristic distances of the ML tree of ORFs in Figure 3a,b and the ML tree of the protein sequences encoded by these ORFs; 149 potyviruses and three rymoviruses are included.

**Figure 5 viruses-12-00132-f005:**
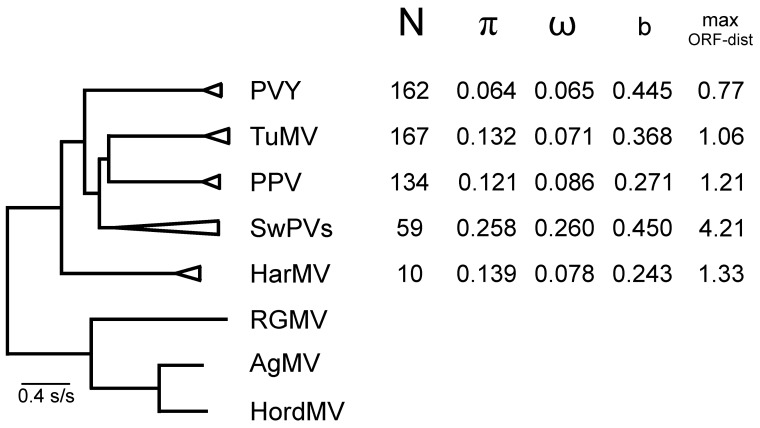
The ML phylogenetic tree (collapsed) of the non-recombinant (n-rec) ORFs of five potyvirus populations. The number of ORF sequences for each virus (N), their average pairwise nt diversity (π) and, for each, the ratio of nonsynonymous nt diversity to synonymous nt diversity ω (dN/dS); also the slope (b) of the linear regression in a graph comparing the pairwise patristic distances in ML trees of the ORF sequences (*y* axis) and encoded polyproteins (*x* axis), and the maximum patristic distance (max ORF-dist) in the patristic distance graph.

**Figure 6 viruses-12-00132-f006:**
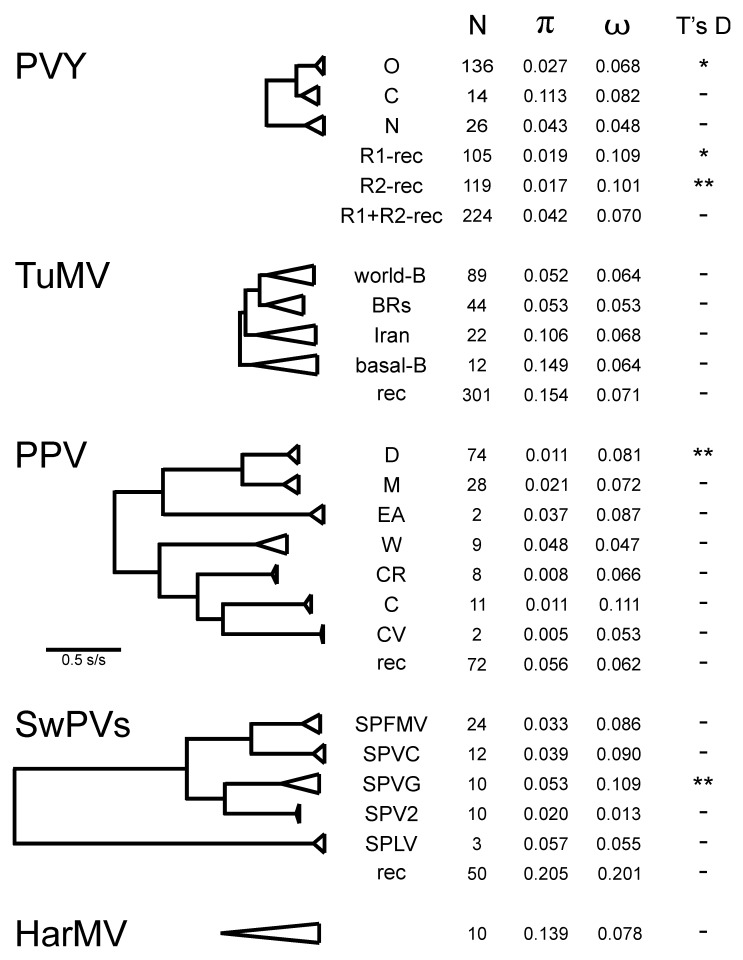
The ML phylogenetic trees (collapsed) of the phylogroups/groups/strains of the non-recombinant (n-rec) ORFs of five potyviruses. The number of ORF sequences for each grouping (N), their average pairwise nt diversity (π), and, for each, the ratio of nonsynonymous nt diversity to synonymous nt diversity ω (dN/dS), and the statistical significance of Tajima’s D coefficient for the group.

**Figure 7 viruses-12-00132-f007:**
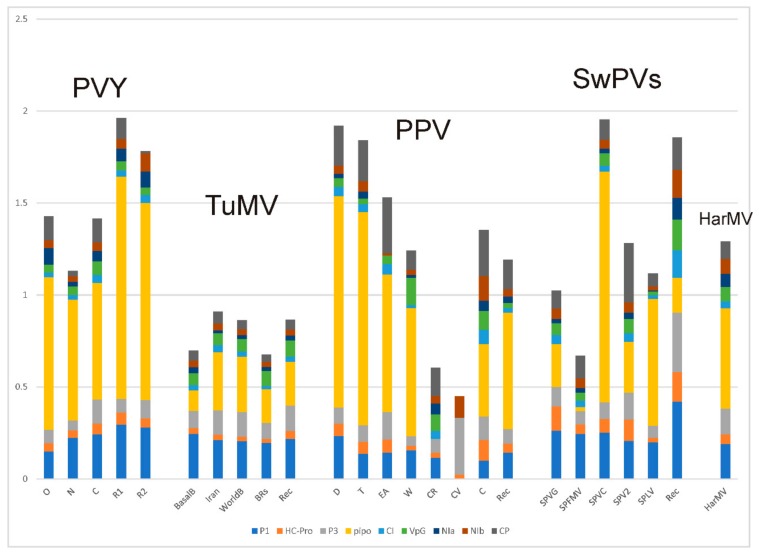
A stacked column chart of the ω (dN/dS) values of the genes of populations of different potyvirus phylogroups/groups/strains. The N-terminal protein (P1) is at the base of each column.

**Table 1 viruses-12-00132-t001:** Timescale analyses of potato virus Y and turnip mosaic virus.

Parameter	PVY			TuMV			
	[105]	[29] and this study	[106]	[100]			
Method	BEAST 1.7.2	BEAST 1.8.2	BEAST 1.8.4	BEAST 1.8.2			
Number of sequences;	28	162	177	106	329	369	369
Sequence length (nucleotides)	9723	8913	564	9432	927	873	855
Region	Polyprotein	Polyprotein	VPg	Polyprotein	Partial HC-Pro	Partial P3	Partial NIb
Sampling date range	1982–2010	1943–2016	1983–2015	1968–2012	1968–2012	1968–2012	1968–2012
Best-fit substitution model	GTR+I+G	GTR+I+Γ_4_	HKY+G_4_	GTR+I+G	GTR+I+G	GTR+I+G	GTR+I+G
Best-fit clock model	UCLD, UCED, SC	UCLD	UCLD	UCED	UCED	UCED	UCED
Best-fit demographic model	ND	EG, BSP	BSP	CZ	CZ	CZ	CZ
TMRCA (95% CI) Year before present (YBP)	161–619 (UCLD), 123–436 (UCED), 525–970 (SC)	1841 (1157–2622) (EG), 1879 (1192–2659) (BSP)	158 (71–269)	1201 (468–2150)	951 (326–1291)	758 (274–1548)	758 (274–1548)
TMRCA effective sample size	ND	238 (EG), 261 (BSP)	ND	ND	ND	ND	ND
Substitution rate (nt/site/year)	Unknown	9.30 × 10^−5^ (6.79 × 10^−5^–1.18 × 10^-4^) (EG), 9.16 × 10^−5^ (6.90 × 10^−5^–1.15 × 10^−4^) (BSP)	5.6 × 10^−4^ (3.35 × 10^−4^–8.17 × 10^-4^)	8.89 × 10^−^^4^ (6.87 × 10^−^^4^–1.30 × 10^−^^3^)	1.41 × 10^−^^3^ (1.09 × 10^−^^3^–1.78 × 10^−^^3^)	1.46 × 10^−^^3^ (1.25 × 10^−^^3^–1.87 × 10^−^^3^)	1.37 × 10^−^^3^ (1.04 × 10^−^^3^–1.73 × 10^−^^3^)
Substitution rates effective sample size	ND	>200	>200	ND	ND	ND	ND
Date (DRT) or cluster-based date (CDRT) randomization test	Not passed	DRT passed	CDRT passed	DRT passed	DRT passed	DRT passed	DRT passed


Best-fit clock model: uncorrelated relaxed lognormal distribution, UCLD; uncorrelated relaxed exponential distribution, UCED; strict clock, SC; expansion growth, EG; Bayesian skyline plot, BSP; constant size, CZ; exponential population growth, EPG. YBP, see sampling date range; ND, not done.

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
