# Peer review of "The Potyviruses: An Evolutionary Synthesis Is Emerging"

_viruses, 2020, doi:10.3390/v12020132_

Round 1
Reviewer 1 Report
General Comments: The manuscript entitled “The potyviruses; an evolutionary synthesis is emerging” submitted to Viruses has great significance. The manuscript is well reviewed and written with clear objective and discussion. Thus, I would like to recommend this manuscript for publication
The manuscript needs some of the minor revision for its improvement, as follows.
The name of viruses should be in italics.
Line 55: potato virus X and potato virus Y in italics.
Line 61: potato virus Y group viruses" will be in italics.
Line 124: The scientific name should always be in italics.
Line 200: turnip mosaic potyvirus should be italics.
Reviewer 2 Report
This paper on potyviruses provides a thorough review of the history of potyvirus research, the distribution of potyviruses in space and time, tries to improve their taxonomy using novel approaches and investigates their relations from an evolutionary perspective.
Although the usage of ω instead of %ID to define species boundary within potyviruses is well supported by the results, it is not clear how this method would perform in case of virus groups other than potyviruses. Nevertheless, even if the method is restricted to potyviruses, it could improve their taxonomy and help to better understand their evolution.
I only require a few minor modifications:
Please remove or replace references to Wikipedia articles. Wikipedia is not a scientific resource with a citable DOI number. Its content is dynamically changing and is edited by non-professionals. Either find a suitable scientific reference or remove the links (most of the cases I found them unnecessary).
Please also remove exclamation marks from the end of some sentences. I think it does not match the style of a scientific paper.
Please write the species names (and other taxonomic names) in italics.
Line 785: remove the word "esoteric".
Author Response
P;lease see attachment

Reviewer 3 Report
Gibbs and coauthors submitted a review on well-known and recently emerged biological and molecular characteristics of Potyviruses, aiming to introduce an evolutionary key to interpretation to potyviral species relationships and differences.
The manuscript is well written and exhaustive. Old and emerging technologies and approaches, such as Computational phylogenetics and metagenomics, are considered and discussed in the light of their advantages and shortcomings.
In my opinion, the present review can be accepted for publication as it is (except for a couple of sentences, see below), and will be of interest to this journal’s readers.
Minor requirements:
Line 264: Figure 7: figures are usually numbered in order of appearance, that is not the case here.
Line 303: Should “104” be removed?
Lines 755-57: this sentence is not clear, please reword
